# The Psychosocial Impacts of Intimate Partner Violence against Women in LMIC Contexts: Toward a Holistic Approach

**DOI:** 10.3390/ijerph192114488

**Published:** 2022-11-04

**Authors:** Michael G. Wessells, Kathleen Kostelny

**Affiliations:** 1Mailman School of Public Health, Columbia University, New York, NY 10032, USA; 2Child Resilience Alliance, 5 River Road, Wilton, CT 06897, USA

**Keywords:** intimate partner violence, intimate partner violence against women, LMIC contexts, psychosocial impact, everyday stressors, holistic approach

## Abstract

Work on the mental health impacts of intimate partner violence in low-and middle-income countries has focused primarily on clinical disorders such as post-traumatic stress disorder, depression, and substance abuse. This paper analyzes how non-clinical, psychosocial impacts from everyday stressors, particularly economic hardships and concern over one’s children, cause extensive suffering and damage women survivors’ well-being, influencing the development and expression of clinical disorders. Using a social ecological framework, the paper analyzes how psychosocial impacts arise at multiple levels, including societal levels where social norms often devalue women and privilege men, and how the stressor accumulation increases the harm caused by intimate partner violence (IPV) against women (IPVAW). Drawing on survivors’ narratives and studies from diverse low and middle income country (LMIC) settings, including armed conflict and natural disaster settings, the paper underscores the importance of understanding both clinical impacts and the non-clinical, psychosocial impacts, which interact with and complement one another. Recognizing the interplay also between IPVAW and other forms of violence against girls and women, the paper calls for a more comprehensive approach to understanding and addressing the impacts of IPVAW. Recognizing the enormous variety within and across countries that are considered to be LMIC settings, the paper cautions against universalized approaches to understanding the effects of IPVAW and helping to support survivors.

## 1. Introduction

Intimate partner violence (IPV) is one of the most widespread and damaging forms of violence worldwide. IPV has been defined as “behavior within an intimate partner relationship that causes physical, sexual or psychological harm, including acts of physical aggression, sexual coercion, psychological abuse and controlling behaviors.” This definition covers violence by both current and former spouses and partners [1]. The World Health Organization (WHO) [1] estimates that IPV affects approximately 30% of girls and women worldwide. Although its exact prevalence remains uncertain due to problems such as underreporting, evidence indicates that it is widespread in Asia and Africa [2,3,4], and Latin America [5], as well as in relatively wealthy countries [6]. Furthermore, IPV is highly associated with humanitarian emergencies due to armed conflict [7], which is prevalent throughout many low and middle income countries (LMICs).

The physical effects of IPV are profound and can include physical injuries, neurological disorders, eating and gastrointestinal disorders, and chronic pain and disease [8,9], among others. IPV can damage sexual and reproductive health by causing problems such as vaginal, anal, or urethral trauma, early and unwanted pregnancy, low birth rate, and HIV and other sexually transmitted infections [1,8,9]. That extreme forms of IPV can cause death is evident in India in the phenomenon of dowry deaths, wherein a husband who is dissatisfied with the dowry payment by his wife’s family murders her, often through hanging, burning, or poisoning [10,11].

The profound mental health impact of IPV is equally horrific and includes Post-Traumatic Stress Disorder (PTSD), depression, suicidal ideation and attempts, anxiety, and substance abuse disorder, among others [8,9,12,13,14,15]. These mental health impacts occur not only in societies in the global North but also in LMIC societies [1,3,16,17,18,19,20,21,22,23] and in adolescent girls as well as in adult women [3]

Any one of these disorders can be debilitating, yet IPV survivors may experience multiple, interacting disorders.

Although these mental disorders are highly important, the impact of IPV on mental health is not captured fully in the symptomatology of mental disorders. Dominant categories of mental disorders do not capture some of the deepest forms of distress, including social distresses that are of profound importance for survivors of IPV. These often include non-clinical, relational forms of distress associated with poverty, poor access to health care, relatively weak systems of law and order, and protection issues such as trafficking, sexual exploitation, and child marriage, among others. As discussed below, these stresses are psychosocial in nature. Because psychosocial and related protection issues tend to be pressing in many LMIC settings, it is important to bring them within the analytic framework for understanding more fully the effects of IPV in LMIC settings.

Focusing on LMIC settings, where the majority of the world’s women live and where women are most likely to experience violence [24], this analytic paper aims to help develop a social ecological analysis of the psychological impacts of IPV that complements the picture provided by clinical analyses. The primary geographic areas of emphasis are Asia and Africa. Although IPV also affects men and people of diverse sexual orientations, the paper focuses on women and girls since they are most severely affected by IPV [1,25]. Reflecting this focus, the paper will refer from here onwards to intimate partner violence against women (IPVAW). Rather than offer a systematic review, the paper draws inductively on literature from diverse LMIC contexts in order to identify important psychosocial dimensions of IPVAW. Section 1 of the paper analyzes how the clinical emphasis on the effects of IPVAW is highly valuable yet provides an individualized, incomplete picture of the psychological and social toll that IPVAW takes on survivors. Next, the paper identifies some key psychosocial dimensions and impacts of IPVAW in LMIC settings at diverse levels of survivors’ social ecologies. The paper concludes with a consideration of the implications for understanding and practice in regard to IPVAW in LMIC settings.

## 2. A Holistic Approach to Mental Health and Psychosocial Well-Being

The clinical approach to IPVAW has been important in calling attention to the powerful mental health impacts of IPVAW and has also helped to guide effective clinical treatment of mental disorders associated with IPVAW. However, important impacts of IPVAW relate also to social suffering and cultural dimensions that are not well captured by a clinical emphasis on mental disorders. These psychosocial impacts complement those identified by the clinical approach yet often do not figure prominently in research oriented discussions of mental health. This section analyzes the importance of taking a holistic, grounded approach that features mental health and psychosocial well-being in analyzing the effects of IPVAW in LMIC settings.

Clinical approaches to mental health have derived primarily from research and practice in Western societies and have been enshrined in standards such as the Diagnostics and Statistical Manuals (DSM) of the American Psychiatric Association and the WHO International Classification of Diseases (ICD). These classifications embody predominantly Western values and approaches that may not fit well with the non-Western social and cultural values and practices of many LMIC countries [26,27,28,29]. For example, clinical approaches are highly individualized since the focus is on maladies and dysfunctions in particular individuals. Individualized approaches, however, are a poor fit with most LMIC societies, which are collectivist in orientation. In LMIC settings, people tend to define suffering and well-being not in individual terms but in terms of groups such as the family, the community, the religious or ethnic group, etc. From this standpoint, the focus on individual disorders provides essential information but does not provide a comprehensive picture of the overall human suffering.

Equally important, LMIC settings differ from Western settings in regard to the prevalence and severity of so called “everyday stressors” such as economic problems and armed conflicts that tear at the fabric of family and community and undermine collective well-being. These everyday stressors are environmental conditions or aspects that can cause psychosocial impacts such as strong worry or non-clinical anxiety. It is fundamentally difficult for people in Western societies to appreciate the pain associated with mass displacement, economic deprivations, or watching one’s children go hungry, die of preventable causes, or miss out on education. In collectivist societies, social stresses such as stigma, loss of social support, and concern for the well-being of one’s children are powerful and are often prioritized by IPVAW survivors, even though they do not themselves require clinical treatment. Evidence from both Western and LMIC contexts indicates that everyday stressors interact with and even mediate the development of full blown clinical syndromes such as PTSD [15,30,31]. Thus they ought to be central to both understanding the impact of IPVAW and steps that are taken to support the well-being of survivors and their families. Collectively, these considerations augur in favor of a more holistic, relational approach to mental health and well-being.

The term “psychosocial” is potentially confusing since it is an adjective that can modify different nouns and has been assigned different meanings by different groups. In the humanitarian field, the term has been an umbrella for diverse social and non-clinical effects and to describe a community-based approach to healing and well-being that emphasizes social mobilization, community empowerment, and the importance of local culture [32,33]. This differs from the terminology in public health, which may categorize as “psychosocial effects” clinical disorders such as depression and PTSD [34,35].

Wanting to stay close to practice in LMIC settings, it is helpful to use the definition offered by Perez, p. 2 [36]: “The word “psychosocial” refers to the dynamic relationship that exists between psychological and social effects, each continually inter-acting with and influencing the other.” The social and psychological dimensions are inextricably interwoven and are socially constructed meanings that people make based on their lived experiences.

## 3. The Psychosocial Impact of IPVAW in Social Ecological Perspective

Understanding the psychosocial impact of IPVAW requires a social ecological perspective [37] that has proven useful in understanding and addressing different forms of gender-based violence [38], including IPV [21]. The ecological framework highlights the systemic nature of IPVAW and can incorporate multi-sectoral stressors such as economic problems, difficult material conditions of living, and poor access to quality health care and education. Although there are diverse categorizations of social levels within an ecological framework, this section examines the psychosocial impacts at individual (survivor), family, community, societal, and international levels. Although these levels are discussed separately, there is extensive interplay between them.

### 3.1. Individual

In LMIC societies, IPVAW survivors will likely perceive individual psychosocial impacts through the lens of relational meaning. For example, a woman survivor who sustained physical injuries may see the impact in terms of her inability to appear in public and sell items that help to support her family. In this respect, the impact may stem not only from her individual physical and emotional pain but also from the suffering associated with her inability to work, be a good mother, and help support her family.

#### 3.1.1. Fear

A pervasive individual impact is fear [2,39,40,41], which in contexts of high intensity IPVAW may be continuous and profound, as has been described as being a “campaign of terror” [42]. A woman who had recently been beaten by her husband stated:

*I was very scared of him. I was terrified of this man*. [39], p. 189. 

The survivors’ fear is not only for their own safety and well-being but also for that of their children [13,43]. The fear is powerful in part because it is ongoing and provokes continuous worry and uncertainty. In the Democratic Republic of Congo (DRC), IPVAW survivors commented that the ongoing, fear-related stress caused such severe fatigue that it left them too weak to care properly for their children [44].

#### 3.1.2. Humiliation, Guilt, Shame, and Stigma

Although discussions of IPVAW frequently highlight physical violence, extensive suffering stems also from psychological violence such as humiliation. Syrian refugees who experienced IPVAW deplored being shouted at and humiliated by their husbands. They felt particularly humiliated when their husbands argued with them or hit them in front of their children. One Syrian woman said:

*I feel like a bird with broken wings. I cannot fly away from home nor can I bear all of this humiliation. Sometimes when he pushes me into the corner I consider running away. It crosses my mind, leaving this whole country. The most important thing is to be hidden from my husband’s sight*.[45], p. 34.

Humiliation also occurs when husbands insult, demean, or shout at their wives in public, a strategy that is often used to demean and isolate the wives. In contexts where polygamy is practiced, husbands may use public humiliations strategically to demonstrate that the wife did not meet her marital obligations, thereby “justifying” his taking of a second wife [44].

Survivors of IPVAW often experience intense guilt since they have been socialized to be the caretakers of relationships and bear responsibility for problems that arise [46]. For example, Vietnamese wives, who are expected to maintain marital harmony [43], report feeling that they are “living in a birdcage” [47], expected to endure such treatment quietly, and feel shame and fear of retaliation that would follow from not living up to that ideal.

In South Africa, too, women report feeling shame over being beaten and psychologically humiliated in the context of their relationship [48]. This shame stems partly from the feelings of guilt and partial responsibility they experienced in navigating complex relationships that included a mixture of love, relational problems, feeling brokenhearted [41], and violence. Such feelings of guilt and shame go hand in hand with self-doubt and diminished self-esteem [13].

Women’s “failure” to fulfill their socially prescribed roles, which demand subjugation, obedience, and maintaining relational harmony, together with the negative status attached to being battered a woman, lead to powerful stigmatization and loss of personal dignity. The combination of stigma and self-blame could cause significant distress, which in LMIC societies, may be amplified by the collectivist orientation and the strength of patriarchal ideologies, norms, and institutionalized power arrangements.

#### 3.1.3. Lost Freedom and Control

As occurs in wealthy countries, male perpetrators in LMIC frequently take control over finances, household decisions, and the behavior of the IPVAW survivor, down to how she dresses, whom she can see, and how she must hide her fear and distress. Locked in relationships in which men use psychological violence, the threat of physical violence, and actual violence to control their partners, women are often unable to work or to go outside and feel isolated.

As a Khmer woman from Cambodia narrated,


*I have a friend who is abused by her husband. She is not allowed to go anywhere. In the dawn, she is ordered to cook foods and bring them on the food tray to him as well. She has to sit down by folding her legs well [proper way for a woman to sit down on the floor in the Khmer custom]. She has to iron his clothes when he is going places. She’s forbidden to finish washing and ironing all clothes…*
 [41], p. 911.

As a result of this overcontrol and loss of agency, IPVAW survivors report feeling helpless [39,41]. This disempowerment of women is particularly concerning in light of the importance of empowerment for well-being [49,50] and human dignity. Furthermore, the loss of agency can impede girls’ and women’s ability to navigate complex environments and negotiate decisions in difficult relationships [3].

Reduced freedom often means that survivors are unable to access needed care, including mental health services. Women may fear that if they go for services, their husbands might respond with more violence. Furthermore, reduced work by survivors may leave them unable to pay the transportation costs for reaching the site of the supports and services. Survivors themselves may decide not to go for services in order to avoid shame or stigma, or to maintain their belief that the abuse was a temporary thing and that holding steady would enable relational harmony.

#### 3.1.4. Social Isolation

A highly significant consequence of IPVAW survivors’ isolation is the loss of social support that normally comes through discussions with friends, people at work, or members of faith communities or civic groups such as women’s groups. Such social support is key in addressing the psychosocial impacts of IPVAW in LMIC settings [22]. The COVID-19 pandemic has only worsened the loss of social support, by leaving large numbers of women, including IPVAW survivors isolated or trapped at home [51].

#### 3.1.5. Stress Reactions

Aside from clinical problems, it is normal for IPVAW survivors to experience strong anxiety, uncertainty, and distress over their own situation and the well-being of their children. Among Syrian women refugees in Lebanon, many of whom experienced IPVAW and various forms of GBV, women said they experienced “daghet” which in Arabic refers to pressure or a feeling of being chronically overwhelmed [40]. It also implies experiencing a strong burden on one’s mental health while retaining the socially acceptable appearance of being strong or unphased [40]. This example illustrates how distress may fit local idioms rather than Western categories [26] and invites further exploration into the lived experiences and subjective yet relational meanings associated with living in contexts of IPVAW.

#### 3.1.6. Substance Abuse

The stresses imposed by IPVAW may lead to increased levels of substance abuse, which itself puts people at risk of harm to their mental health and psychosocial well-being [1]. In a study of females aged 13–24 years in Colombia, people who experienced IPVAW were more likely than people who did not experience IPVAW to report that they had used drugs in the past 30 days. This substance abuse was harmful in its own right, and it also contributed to work absenteeism and reduced labor productivity [52].

#### 3.1.7. Economic Effects

IPVAW episodes have significant direct and indirect economic impacts [52,53,54,55]. Direct impacts often relate to the costs for health treatment. A survivor who seeks health care following an IPVAW episode in a LMIC setting incurs out-of-pocket expenditures that range from US $29.72 in South Africa to US $156.11 in Romania [54]. Accumulated direct costs are likely to be significantly higher due to repeated violent episodes, the occurrence of multiple injuries, and the exacerbation of old injuries.

The indirect costs of episodes of IPVAW include income loss due to survivors missing or cutting back on paid work due to injuries, shame, or fear, or a combination of these. Affected by fear, anxiety, and uncertainty, IPVAW survivors may also have challenges in concentrating and may experience reduced productivity, which can lead to income reductions. These income reductions, which range across diverse settings from $73.84 to $2151.48 per IPVAW episode [54], sharply increase economic stresses. Girls who experience IPVAW may suffer educational losses such as time out of school, difficulty learning, performance decrements in school, and early dropout [56,57]. These educational challenges can deprive girls of the long-term economic benefits associated with education in LMIC settings [58].

#### 3.1.8. Enduring Suffering

For many IPVAW survivors, the stresses discussed above are continuous and often long-term, though they often hide their pain from others. As two African immigrant women put it:

*…Whenever I feel that my husband has treated me badly, may be he screamed at me or something, I just keep calm instead of giving it back to him, I always keep quiet. But I realise that all these things (IPV) have negative impact on a women’s health*.

*For example, if your husband has just beaten you mercilessly and someone knocks at your door, you pretend as if you are the sweetest couple in the whole world*.[59], p. 1661.

If survivors disclosed their pain to other people, this could shame the family or anger the husband, leading to further IPVAW.

Depending on the context, survivors may decide to stay with their abusive partners owing to financial uncertainty, which frequently relates to concerns about their children’s well-being, or the dangers associated with being on their own [60]. In Middle Eastern societies, survivors’ choices are often influenced by the strong stigma associated with divorce, the lack of family support for leaving, and fear of losing custody of their children [60,61,62,63]. Although survivors take steps such as praying and keeping silent as means of minimizing conflict and making their situation more bearable [61], the burden of ongoing distress remains considerable and prolonged.

### 3.2. Family

In LMIC settings, the family level is particularly important to consider in analyzing the psychosocial impact of IPVAW. In most LMIC settings, marriage is the primary venue for intimate relationships and for legitimacy within them. Usually, marriage and relationships are seen as family matters rather than individual or couple matters only.

#### 3.2.1. Family Violence

IPVAW and its psychosocial effects are grounded in wider, socially sanctioned, systems of abuse and mistreatment of women. In both Asia and sub-Saharan Africa, marital rape and sexual degradation is widespread and is often depicted in customary law as a husband’s prerogative [64,65]. Widespread acceptance of marital rape and the absence of or failures to enforce laws against rape inside the family can leave women feeling fearful and without protection.

Similar dynamics are at play in dowry practices that are widespread in Asia, northern Africa, and other regions [66]. In countries such as India, Pakistan, or Bangladesh, for example, the bride’s family is expected to pay the groom and his family a substantial amount, often in forms such as livestock or property, at the time of marriage. If economic circumstances worsen or other challenges arise, there may be delays in or failures to pay the dowry. This leads to extreme pressures being placed on the bride, with resulting increased rates of suicide among recent brides [10]. In countries such as India, family members have sometimes engaged in dowry killing [10], despite laws against this horrendous form of murder.

In Muslim societies such as Afghanistan and Pakistan, there may be fear of honor killing, usually by a family member [2]. In sub-Saharan Africa, family members often play a role in enabling harmful practices such as female genital mutilation/cutting [67].

#### 3.2.2. Effects on Intimate Relationships

In countries such as Afghanistan, quarreling often increases in intensity, creating a conflict spiral that often culminates in IPVAW [68]. In areas such as sub-Saharan Africa, the conflict spiral and IPVAW often occur in association with the excessive drinking by the male partner [44,69]. An IPVAW episode can transform intimate relationships and often sparks additional conflict, mutual anger and recriminations, uncertainty, and increased efforts by the perpetrator to control his partner by physical and psychological means. These have significant psychosocial impacts on survivors, as discussed above.

The psychosocial impact of IPVAW on the relationship, however, owes in no small part to the attributions that the woman makes about what caused the violence. If, for example, a woman whose husband has hit her sees the violence as situational rather than dispositional, she might be inclined to view the violence as temporary and believe that he will change. As one South African woman stated,

*Every single time I thought: ‘Ok, maybe, maybe he’s just going through a bad patch, you know he will change, he will change.’ He never*.[39], p. 193.

However, if the woman experiences some guilt for the violent transaction, she may feel she has let her relationship and her family down [39]. Such guilt may lead survivors to continue enduring the damaging relationship, even when there is no realistic hope of significant relational improvement.

The effects of IPVAW on one relationship can spill over into future intimate partner relationships. For example, adolescent girls who have been affected by IPVAW are more likely to engage in the future in risky sexual behavior, unwanted pregnancies, or substance abuse [9,52]. This behavior could lead them into relationships that are not founded on mutual love and caring and that may become spaces for physical, sexual, and psychological abuse.

#### 3.2.3. Polygamy

Polygamy is widespread throughout sub-Saharan Africa [70,71] and in diverse parts of the world. In general, polygamous relationships have higher rates of IPVAW than do monogamous relationships [14,71,72,73]. Furthermore, women in polygamous relationships are more likely to approve of spouse beating [72].

The higher levels of IPVAW in polygamous relationships engender significant psychosocial distress. In addition, much distress stems from competition [74] and even hostility between the co-wives [75,76]. Traditionally, the first wife, who is usually the oldest, has authority over other wives and may treat them in cruel, overbearing ways [77]. Although senior and junior wives may collaborate, senior wives usually experience fear, anger, sadness, and loss when the husband takes on a junior wife. The felt competition is a struggle not only for status but also for the economic resources that strongly influence the well-being of the woman and her children. Stresses also come from the husband’s emotional or sexual favoritism [78].

In this system of competition and jealousy, conflicts between wives may escalate and create difficulties for the husband, who responds to the ‘bad behavior’ of a wife (often the senior co-wife) with violence. Psychological violence may also become an endemic part of the co-wives’ relationships. Fortunately, this does not happen in all polygamous relationships or in polygamous relationships in all countries [71].

#### 3.2.4. Intergenerational Effects

Current research estimates that nearly one third of children in LMIC contexts have been exposed to IPVAW in their lifetime [79]. Exposure may occur through children’s direct observation of IPVAW episodes, and it may also occur by children becoming aware of the IPVAW, for example by seeing their mothers’ wounds, overhearing discussions about IPVIPVAW episodes, or experiencing life changes as a consequence of the violence [80].

Exposure to IPVAW can have powerful effects on children’s mental health and psychosocial well-being. It can lead to diverse mental health problems such as mood and anxiety disorders such as PTSD [81,82,83] and also to conduct disorders and adjustment problems [84,85,86,87,88]. In Democratic Republic of Congo, exposure to IPVAW increased stigma, especially for girls, and led to missed days of school and externalizing behavior [89]. As a 10-year-old girl from Rwanda put it:


*I would come home and find Daddy beating up my mom. I would run away crying and go back to school without eating. I was very worried about my mother, thinking that when I came home, I would find her dead. I couldn’t concentrate to study... I failed the second year. Today I’m doing very well in school because my father doesn’t beat my mother anymore…*
[90], p. 45.

Exposure to IPVAW can lead to substance abuse and risky behavior such as unsafe sex [91], and is associated with long-term health problems such as cardiovascular disease, cancer, diabetes, and reproductive health problems [92,93].

In addition, children who have witnessed IPVAW are at increased risk for perpetrating or experiencing IPVAW as adults [94,95,96]. One pathway for intergenerational transmission of IPVAW is that children learn to see a husband’s violence against his wife as normal.


*I um grew up with my parents and that was also like my father was also drinking and there I also witnessed like abuse in the family… My father used to hit my mother and all that stuff. But, um in the end actually we grew up believing that it was right. That the husband must hit the wife, that is how we grew up…*
[39], p. 195.

Exposure to IPVAW may have strong psychosocial effects on children by shattering the safety of their home and family. Hearing episodes of IPVAW or seeing their mothers’ wounds may evoke fear and anxiety for their mother’s well-being. Experiencing their mother’s fear and reduced ability to care for them, children may worry for their own well-being and harbor uncertainty about their future. In some contexts, children may fear that they will be separated from their families, which may be deemed by authorities to be unsafe for children. Thinking that they must have done something wrong that brought about the mother’s beating, children may also experience guilt and loss of self-esteem. Due to household economic pressures, children may have to drop out of school when they see education as their source of hope for the future, and they may engage in dangerous forms of labor or work that puts them at risk of exploitation and sexual violence [97]. Even if they stay in school, their learning opportunities may be reduced, as children’s exposure to IPVAW is associated with cognitive impairment and reduced educational achievement [98].

These effects on children add to the psychosocial distress of mothers, who in many LMIC settings, define their well-being in no small part based on the well-being of their children [99] and their effectiveness as mothers. Seeing the negative impact on their children only adds to mothers’ burden of guilt and shame, and undermines their greatest hope, which is to be a good mother.

#### 3.2.5. Lack of Family Support

In many LMIC contexts, married women look to their parents and extended families for support in difficult times. Yet, in settings as diverse as Colombia [100], northern Uganda [101], or Palestine [102], IPVAW survivors do not receive emotional or other support from their families, especially in regard to divorce. This lack of support only adds to survivors’ isolation, hopelessness, powerlessness, and lack of protection.

### 3.3. Community

Communities are often sites of perpetration of IPVAW and other forms of violence against girls and women, and seldom offer the needed supports and protections for IPVAW survivors. The discussion below considers non-conflict settings and also conflict settings, since armed conflict is not only widespread in LMIC settings but also enables more suffering related to IPVAW.

#### 3.3.1. Non-Conflict Settings

Although IPVAW is prohibited by law in most countries, it is frequently normalized and seen as natural according to local customs. Further, community norms may require men to “punish” disobedient wives using violence, and the failure to do so can evoke criticism. Having grown up observing IPVAW in their own families and learning from one another, men in a community may accept the practice, blame the victim when IPVAW occurs, and do relatively little to intervene against it. Community norms are usually not to intervene unless the husband and wife ask for help, which is seldom done by self-righteous, angry perpetrators. This situation usually leaves women feeling fearful, uncertain about the future, and powerless to stop the IPVAW.

In some settings, horrendous forms of IPVAW, including murder, occur in public spaces yet without concerted community outrage or intervention. In 2022 in northern Egypt, for example, a female university student who sought a career rather than a traditional marriage was fatally stabbed by a man who had asked to marry her several times but had been refused [103]. Although this horrific murder evoked calls for legal protections and human rights for women, some social media responses blamed the victim for not having covered herself properly to avoid tempting men [103].

Such incidents are part of a much wider pattern of violence against girls and women, which may include sexual harassment, rape, sexual exploitation, early marriage, teenage pregnancy, and female genital mutilation/cutting. This extensive, normalized violence against women and girls in public spaces, often with impunity or even community support, engenders continuous fear and violates the dignity of women and girls. This fear may lead to withdrawal from public places, severely limiting survivors’ freedom, agency, career aspirations, ability to work, and social connections that are key for well-being. Complementing this fear are the stigma, victim blaming, shame, and ongoing discrimination that women often face at community level in LMIC contexts.

This constellation of factors may severely limit survivors’ ability to work, and the resulting income loss can make it difficult for survivors to support their children, which is a profound source of distress. The mother’s worry and guilt will likely be particularly strong if her children have to drop out of school. Mothers may also worry about the stigma extending to her children, as a woman’s disobedience to her husband is often perceived as a stain on her family.

Distress stems also from the paucity of and problems accessing community supports and services for IPVAW survivors. In many areas, they have no safe place to go and no quality mental health services to turn to. Cut off from their friends and natural helpers in the community, they may fear seeking services or supports since the husband might see going for treatment as another transgression requiring even harsher “punishment.” Furthermore, feeling the weight of stigma and economic losses, survivors might elect not to seek services because of their social or economic costs.

Adding to the distress is the powerlessness to report IPVAW that women in some cultures experience. In Arab cultures, for example, family violence is seen as a family secret that must be kept within the family [45]. Furthermore, cultural values that demand the woman’s obedience to her husband, as well as her financial dependency on him, can make it difficult for women to report their victimization [45,104]. These challenges in reporting enable impunity and undermine efforts to protect survivors.

#### 3.3.2. Conflict-Affected Settings

Evidence from countries such as Colombia [52], DRC [105], and Northern Uganda [101,106] indicates that armed conflict is associated with higher rates of IPVAW. Countries such as these that have complex humanitarian emergencies also have pervasive gender-based violence [7], severe economic problems, and a breakdown of law and order. Even if people manage to flee their country of origin, becoming refugees, a very difficult psychosocial situation may occur. For example, Syrian refugees living in Jordan may live in contexts that have law and order, yet they suffer the twin effects of social isolation and discrimination [45].

In addition, armed conflict can increase the severity of IPVAW. In South Sudan, which has been torn by long-term armed conflict that has devastated the economy, displaced large numbers of people, and undermined the rule of law, women reported that IPVAW has become more brutal. As one survivor put it:

*Before the crisis, we were fighting. Now they are removing our eyes; they are kicking us in the stomach*.[17], p. 17.

Women also reported that the frequency of IPVAW had also increased. [17]

The increasing IPVAW in South Sudan and other war torn countries stems also from harsher economic conditions. In South Sudan, a man supplies a “bride price,” usually in the form of cows, to the family of a girl in order to marry her. Due to very harsh economic conditions, families have married their daughters at younger ages as a strategy for regaining cattle or obtaining wealth, leading to increased early marriage. The inability of men who cannot pay the bride price has also led to an increase in abductions of girls.

*Abduction of young girls occurs in order to take them as wife because men have no money/cows for dowry*.[17], p. 17.

The psychosocial impacts of this system are both powerful and diverse. Although girls may want to help support their families, they may feel exploited by being married off as a means of increasing family wealth. To be a tradeable commodity is severely dehumanizing and devaluing. Girls’ stress is increased by the possibility that they might be given to a particularly abusive man or in a polygamous relationship that is saturated with jealousies, suspicions, and frequent beatings. Knowing that the IPVAW episodes are increasingly brutal leaves young women trapped in fear for their own safety and the well-being of their children. Compounding this fear related to IPVAW are the fears associated with the breakdown of law and order, which enables increasingly widespread crimes such as rape and cattle raiding.

*People who carry guns here, not soldiers, are causing more violence in our community. They are the ones raiding cattle, stealing other people’s properties, raping women and girls, and creating insecurity at the borders and in the bush*.[17], p. 41.

By normalizing violence against women, this system of violence leaves girls and women living in continuous fear. It also enables ongoing violence at community level that further erodes the safety and well-being of everyone and imposes extensive costs on the community through lost productivity, reduced social cohesion, and the withdrawal of women, who are key resources, from community life.

Similarly, armed conflict can shatter or undermine the services and supports that IPVAW survivors need. In many conflict-affected areas, the necessary health, legal, mental health and psychosocial supports (MHPSS) and other services and supports are not available. When they are available, the service providers may be severely stressed by overwork and the emotional, social, and economic burdens of war, thereby weakening key resources in the community [21]. The same pattern likely applies to non-formal supports such as the psychosocial supports that might ordinarily be provided by friends, neighbors, teachers, women’s groups, and religious groups. In sub-Saharan Africa, IPVAW survivors often try to turn to community leaders and non-formal community processes for help. Unfortunately, armed conflict frequently weakens such community systems, thereby weakening non-formal supports for survivors [107]. The weakening or depletion of both formal and natural helping resources in the community undermines community support for IPVAW survivors and likely makes communities seem less caring. The lack of support adds to psychosocial distress, as survivors feel they have no one to turn to for help.

### 3.4. Societal

At societal level, patriarchal structures and systemic discrimination privilege men, institutionalize inequities between women and men, and enable IPVAW. In many respects, a circular relationship exists between structural violence against girls and women and physical violence against them [108]. This section first examines how patriarchy supports IPVAW, and then discusses how it enables wider violence against girls and women, with the combined effect of normalizing violence and enabling further IPVAW.

#### 3.4.1. Patriarchy and IPVAW

Undergirding IPVAW are cultural norms and ingrained beliefs and practices that treat men as heads of households who are expected to punish disobedient wives with violence and that may regard women as a form of property. Societal norms demand that women obey their husbands, thereby setting the stage for serious conflict when women refuse to do so. Beatings, humiliations and other forms of violence are used to put the woman in her place and ensure future obedience, thereby reasserting male dominance and supporting the system of male privilege and structural violence. Often such practices are supported by religious doctrine [38]. Victim-blaming is inherent in this system since women are viewed as having brought punishment and shame on themselves.

Although the particulars vary significantly by region, examples of these dynamics are found in LMIC settings worldwide. Throughout sub-Saharan Africa, men are regarded as undisputed heads of household, with women subjugated to a secondary status and expected to obey male authority [39]. A woman’s disobedience to her husband is to be answered by violence, which is often euphemized as “punishment” designed in part to teach her the proper obedience and behavior. Particularly severe beating may be meted out for infidelity. This transgenerational practice is so enshrined in social norms that a husband will be criticized harshly by extended family and community members if he fails to properly discipline his disobedient or unfaithful wife. As stated by a Rwandan man,

*…I grew up listening to my father tell my mother that under no circumstances will he be “inganzwa” (a man submissive to his wife). To get respect, you had to hit her. My mother was submissive while my father was the head of our family. He made all the family decisions; my mother had no say in them. My father often beat my mother and she prevented us from telling other people. We grew up in this degrading atmosphere. Later, when I started my home, I would beat my wife to get her to respect me, as my father had set an example for me. I was the one who had to manage everything and my wife was of no value to me. I didn’t want her to leave my house and I wouldn’t let her talk to others. When she took refuge with her parents, my mother-in-law would tell her to go home immediately to avoid the family shame if she stayed, because “this is how the household is built” (“subira iwawe, niko zubakwa”)*.[90], p. 25.

Within this patriarchal system, men decide what counts as “disobedience.” In practice, a husband may beat his wife severely for infidelity on the basis of his suspicion rather than for actual infidelity.

In Arabic speaking countries, too, women are expected to be demure and obedient to their husbands, who use violence against wives and women who do not adhere to the norm [109]. In Latin America and the Caribbean, people in rural areas and in disadvantaged groups tend to have favorable attitudes toward the use of IPVAW by a male in response to infidelity by his partner [110]. Such norms also exist in Asian countries [47,66].

Some of the most violent norms occur in India and Pakistan, where men seek to protect family and community honor by insuring that their daughters adhere to patriarchal ideas about what counts as “good behavior” by a woman. If a woman departs from the expected behavior, she is seen as having undermined the honor of her husband, who is expected to control her and hold her responsible for maintaining the highest standard of behavior. Viewed as a major transgression, behavior such as adultery or sex outside of wedlock is seen as causing loss of honor that “justifies” serious punishment, including killing [111].

Together with wider patterns of violence against women and girls, these social norms and widespread IPVAW serve to normalize IPVAW and violence against girls and women. Although this normalization frequently goes unspoken, it serves to reduce societal outrage against and responsiveness to violence against girls and women. Quietly, normalization can create the veneer of legitimacy by making IPVAW seem to be the natural state of affairs.

Although much remains to be learned about the societal costs of IPVAW, evidence from Colombia indicates that physical IPVAW imposed health costs of $90.6 million [52]. In a country struggling to recover from decades of armed conflict, this staggering figure likely impedes positive development and detracts from the country’s ability to provide supports and services for the neediest people, including IPVAW survivors.

#### 3.4.2. Patriarchal Systems and Wider Violence against Girls and Women

Beyond the family, patriarchal systems privilege men and subjugate women. Massive discrimination in education, employment, public office holding, and other domains relegate women to a secondary status. This system of male domination makes it seem “natural” that men hold the power and use it to dominate women. Patriarchal systems support various forms of violence, such as rape and sexual exploitation, which carry increased risk of HIV and AIDS as well as other STIs, and may lead to early pregnancy, which is associated with a welter of health risks. These forms of violence are even more extensive and profound in conflict torn societies [7]. Following norms of victim blaming, women who have suffered violence that they did nothing to provoke may be criticized for having dressed or behaved in a provocative manner. A rape survivor in DRC, for example, may suffer extensively owing to stigma and also abandonment by her husband [112].

Patriarchal systems demand and support the subjugation, discrimination, and violence against women and girls; relegates women and girls to secondary status; and robs them of a sense of having equal value and opportunity. Furthermore, many women are socialized in a manner that leads them to internalize a sense of their own inferiority. In countries where girls are given in marriage by their families, girls and young women may feel that they are a form of ‘property’ and have few rights of their own. As a South Sudanese man stated,

*The women are inheritable when husbands pass away. The next of kin or brother of her husband takes her to be a wife without her consent. This affects most women psychologically and gives them mental illness. She may be tortured by the next of kin or her husband’s brother*.[17], p. 18.

At the same time, societal discrimination regarding land ownership and property takes a heavy psychosocial toll on women in regions such as sub-Saharan Africa. In Zimbabwe, older women who have lost their husbands are frequently mistreated by his family, who may insult her on a continuing basis, deny her food, or take over her former husband’s farming area, thereby denying her a livelihood [113]. They may also take over her former husband’s property, as many laws and practices deny land ownership to women. In northern Uganda, during its multi-decade armed conflict, many men died, yet their wives suffered extensively since they were not entitled to inherit their husbands’ land [114], which was badly needed for farming and supporting their children. In some parts of Kenya, in addition to not being able to inherit their husband’s land, wives are also forced to give up their children to their husband’s relatives.

Societal discrimination against women and girls in LMIC contexts frequently leads to girls and women having lower levels of education than men, often with inappropriate “justifications” such as the view that “women’s place is in the home” so they do not need education as much as men do. As illustrated by the case of Malala Yousafzai in Pakistan [115] education discrimination in a highly conservative regime can be backed by use of physical violence. Aside from physical violence, education discrimination is a form of psychological violence that damages girls’ hopes since they often have strong a desire to continue their education and see education as their pathway to hope and better life [116]). Discrimination against women occurs in diverse realms, as women in LMIC settings are typically underrepresented in the highest leadership positions, such as President of a country, CEO of a company, or Chair of the Board for major corporations.

This discrimination is a form of slow drip psychological violence that imposes daily indignities and suffering associated with being treated as a second-class citizen while simultaneously watching men and boys enjoying unfair advantages. Augmenting the psychological violence are frequent micro-aggressions against women and girls [117,118]. Micro-aggressions may involve seemingly small slights such as a patient at a health clinic calling a female in a white coat “Nurse” when she in fact is the attending physician, or a male commenting more on a co-worker’s looks than her competencies. Such degrading and demeaning treatment can cause psychosocial harm, which only increases as the micro-aggressions accumulate.

Important as well are the psychosocial impacts on society. Patriarchy, extensive IPVAW, and other forms of GBV collectively sow deep divisions within a society, weaken social cohesion, and help to normalize violence against half its population. They also damage the women and girls who are key family and community resources, leaders, and enablers of development [119]. In most societies, women are key caregivers and peacebuilders who help to strengthen norms of interdependence. The assault on women as societal resources, coupled with the weakening of development and the embrace of violence, amounts to powerful psychosocial loss for society.

### 3.5. International

As armed conflicts and political turmoil increase worldwide, increasing numbers of people are refugees and migrants to other countries who experience increased stressors and reduced social controls that may contribute to IPVAW. Current evidence indicates that being a refugee increases the risks of IPVAW [107,120]; makes the IPVAW impacts more severe [121,122,123] and decreases the likelihood that survivors will receive the services and supports they need and are entitled to [107]. Further, the perpetrators may worsen the psychosocial impact on the survivors by threatening to report them to the immigration authorities [121,123].

For example, Colombian women who live in Ecuador near its border with Colombia frequently have a violent life course, including violence experienced during their movement across the border, and face severe stresses such as not having proper documentation, economic deprivations, and loss of their support networks [121]. Many of their husbands engage in IPVAW and use their wives’ lack of social networks to strengthen the isolation of and control over their wives. Living in a different society and lacking documentation or access to close friends, the women survivors have nowhere to turn and do not dare to report the IPVAW to authorities out of fear that they will be imprisoned or deported [121].

Refugee survivors’ psychosocial well-being is compromised not only by the IPVAW but also by being in a new country which has a different language and culture, and which does not include protection or accessible supports that the survivors will actually use. A study of refugees living in camps in South Sudan, Kenya, and Iraqi Kurdistan found that IPVAW was widespread among the refugees but IPVAW survivors had little protection from their abusive husbands, and engaged formal IPVAW responses (e.g., through a UN agency or police) only as a last resort [60]. Furthermore, the customary supports for the refugees, who lived in camps, had been weakened by armed conflict and were seldom effective. For example, if an IPVAW survivor went to block leaders, who were usually men, block leaders had no significant power since they could not fine the perpetrator or decide upon a divorce.

Similar problems arise for migrant women who are affected by IPVAW. Like refugees, they frequently suffer IPVAW and other forms of GBV, have problems with their legal status, face significant economic hardships, and have to navigate a different culture, a new language, and a loss of social networks and supports [123,124]. As climate change increases and as economic hardships and inequities spread, the impacts of IPVAW on migrant girls and women will likely increase significantly in the coming decades, only adding to international violence against women and girls.

## 4. Discussion and Conclusions

The psychosocial effects of IPVAW discussed above have notable similarities with those seen in Western contexts such as fear, shame, economic hardship, stigma, social isolation, and concern for one’s children. However, important differences are also evident. In LMIC contexts, the economic stresses imposed by IPVAW, which can mean that one’s children go unfed today, are fundamentally greater on average than those seen in most Western contexts. Furthermore, health, protection, and MHPSS services and supports tend to be less widely available in LMIC countries, particularly in rural areas. The concentration of armed conflicts in LMIC areas exacerbates the problems of economics and lack of services and supports, and coupled with a breakdown of law and order and protections for women and girls, it increases the spread and severity of IPVAW. LMIC countries may also have cultural practices such as dowry deaths and honor killings of women [10] that are more widespread than they are in Western countries.The evidence discussed above indicates that across diverse LMIC contexts, with a primary emphasis on Asia and Africa, IPVAW generates pervasive and powerful psychosocial distress at all levels of survivors’ social ecologies. Although this paper has discussed psychosocial stresses separately for purposes of clarity, it is important to recognize an individual survivor usually is affected by multiple, interacting psychosocial stressors. At individual level, for example, survivors may experience fear, humiliation, guilt, and concern about their children at the same time. This stressor or risk accumulation can lead to a sharp increase in the likelihood of negative developmental outcomes, suffering, and psychopathology [125,126]. Risk accumulation also occurs across ecological levels. The psychosocial suffering of a woman survivor may be increased by a combination of factors such as individual mistreatment, coupled with mistreatment by her extended family, isolation from her friends and peers, and stigma at community level. The accumulation of stressors and the combined impact on IPVAW survivors indicates the importance of addressing the psychosocial impacts of IPVAW.

In addition, risk accumulation contributes to chronic stress that can produce long-term damage to health [127]. For example, children’s early exposure to multiple and chronic stress, as often occurs in IPVAW settings that combine exposure to violence with economic problems and stigma, can enable life-long health problems such as cardiovascular disease, diabetes, and respiratory disease, as well as unhealthy life styles. For women survivors of IPVAW, the accumulated stress can add to the physical health damage already caused by the IPVAW, and can add to her concerns over her children’s well-being if her own health declines.

These multi-level, multi-systemic effects caution against excessive emphasis on the provision of individualized supports for IPVAW survivors [43]. This distress and stressor accumulation causes extensive harm and warrants systematic attention and steps toward its prevention and alleviation. Survivors’ narratives frequently prioritize and highlight the importance of non-clinical stresses such as concern for the well-being of one’s children. Since survivors’ psychosocial suffering is profound and may contribute to the development and expression of clinical disorders [31], it is appropriate to speak not of mental health alone but of “mental health and psychosocial well-being,” as has been done in the global guidelines on GBV [128].

### 4.1. Toward a Holistic Approach

It is time to bring psychosocial distress out of the margins and make it central in research, practice, and policy analyses. The increased attention to psychosocial distress should complement the frequently seen emphasis on mental disorders. Research on psychosocial impacts of IPVAW in LMIC settings is in its early stages and needs to become as widespread as is the focus on the mental health impacts of IPVAW. Future research is needed to examine the interplay between mental health and economic distress, distress associated with the possibility of losing one’s children, and profound, ongoing fear, among others. In practice, strengthening the capacities for the treatment of mental disorders remains a high priority. However, equal attention should be given to alleviating and preventing psychosocial distress, including multi-sectoral distress from factors such as economic difficulties, social isolation, poor access to health care or education, and sub-standard housing. Practical supports should strengthen the survivors’ agency, the undermining of which is a central impact of IPVAW. Among donors and policy leaders, too, a more holistic approach that systematically integrates mental health and psychosocial supports is highly needed in the humanitarian and development arenas in LMIC settings. Efforts to scale up clinical treatments for mental disorders should not be implemented in a fragmented manner but should be complemented by efforts of equal intensity to enable the psychosocial supports that are of highest priority to the IPVAW survivors in a particular context.

To be effective, a holistic approach to supporting IPVAW survivors must attend carefully to the context. The term “LMIC settings” can imply a homogeneity that does not exist. The literature discussed above indicates that extensive diversity exists among LMIC settings with respect to cultural beliefs, norms, and practices; conflict vs. non-conflict settings; economic conditions; and political and social conditions. This diversity cautions against “one size fits all” approaches to addressing psychosocial distress. Efforts to address psychosocial distress will need to listen carefully to survivors, learn about the particulars of the context and the meanings associated with them, and build on local strengths and resources that are meaningful to survivors. As contextually relevant supports are implemented, a high priority is to document and evaluate their effectiveness, as psychosocial support needs a stronger evidence base [129].

A holistic approach must also have an ecological orientation that attends to stressor accumulation within and across levels. The establishment of supports at one level, such as the community level, can be valuable but will be limited unless complementary, coordinated supports are established at other levels such as individual and family levels. In fact, there are useful models of such multi-level supports (e.g., the SASA! Model; see [130]. From an MHPSS perspective, a key is to interweave individual supports with valued social supports that are meaningful to survivors and potentially sustainable. An ecological orientation should also include intentional efforts to help transform the societal discrimination against women and social norms that undergird IPVAW [38,108].

### 4.2. Limitations

This paper has intentionally focused on psychosocial stresses and risks in order to provide a more comprehensive analysis of the negative impact of IPVAW. This analytic focus, however, should not obscure the significant resilience demonstrated by many IPVAW survivors. Survivors make many complex decisions and draw on available strengths and resources as they attempt to survive, navigate, and make meaning in complex relationships and circumstances associated with IPVAW [48,75] strengths based approach is particularly important in thinking about conceptualizing and developing practical supports for both response and prevention [2].

This paper’s analysis of the psychosocial impacts of IPVAW in LMIC contexts is best regarded as preliminary. Research on IPVAW in LMIC contexts is in its early stages, and much of the research done on psychological impacts has focused on clinical disorders. Further, psychosocial impacts often include culturally constructed dimensions, making it challenging to generalize across different contexts. The significant diversity that exists within the rubric “LMIC settings” resists efforts at generalization, particularly at a moment when much remains to be learned about IPVAW in different contexts and the divergent sub-groups that may exist within a particular country or context. This qualification is particularly important since this paper emphasized Africa and Asia. Additional work needs to analyze the psychosocial impact of IPVAW in Latin America and other geographic areas that include LMIC settings. Because evidence regarding psychosocial well-being is still relatively weak and many cultures have not yet been studied, it is important to approach the tasks of learning about IPVAW and supporting survivors in different settings with cultural humility.

## Data Availability

Not applicable.

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
