# Peer review of "The Psychosocial Impacts of Intimate Partner Violence against Women in LMIC Contexts: Toward a Holistic Approach"

_ijerph, 2022, doi:10.3390/ijerph192114488_

Round 1

Reviewer 1 Report

This paper discusses an important topic - Intimate Partner Violence (IPV), which is an old, but growing social problem at global level according to recent statistics. The theoretical holistic approach of this complex problem through the social ecological framework is highly relevant, being able to explain the complex interplay of both causes and effects of IPV. It is not clear if the paper claims to be research based or a literature review. In the first case, there is no empirical data collected by the authors; in the second case the review includes relevant articles and other literature reviews but does not use any design specific for a systematic literature review. The suggestion is: to either report on new empirical date or to use a systematic review methodology to justify the selected papers on which the analysis is made. Then, the findings could follow the structure of subtitle 3. The Psychosocial Impact of IPV in Social Ecological Perspective. The section 3.6 Accumulation and Interaction is key for the argument of this paper and should be included in the Discussion section and be more detailed.

Another weakness is the use of low- and middle-income countries (LMIC) concept, with examples mainly from Asia and Africa, very few from Latin America (p. 12, 14) and just one mention on a EU country (Romania, on p. 6 -  “economic effects”).  The suggestion is either to clarify from the beginning that the examples are drawn from LMIC contexts from these geographic areas (which have many different cultural specificities compared to EU LMIC) or to broaden the literature used referring to LMIC countries globally. Even the limits section does specify that there is no homogeneity between the LMIC contexts and argues for “cultural humility”, the structure of the paper and the way the data are presented as relevant for the LMIC contexts implies some degree of generalization, outside the regions presented as examples. Please specify the geographical regions to draw upon the findings.

To conclude, the paper argument for a holistic approach and broader focus on psychosocial well-being, compared to the limiting focus on mental health is valuable, but the design/ methodology of the paper needs major revisions.

Author Response

Reviewer 1 raised several concerns and also offered valuable suggestions which are addressed here point by point.

Methodology. Noting that it is not clear whether the paper claims to be research based or a literature review, Reviewer 1 suggested that we provide either new empirical data or a systematic review of the literature. Since neither of the suggested approaches are what is most needed in the current state of knowledge, we respectfully decided that it is not possible to follow this suggestion, as justified below.

A key part our paper is conceptual and analytic, and aims to broaden the definition of “psychosocial” impacts of IPV. At present, the term “psychosocial” is contested and used in ways that are contradictory. In global health, for example, papers that examine “psychosocial impacts of IPV” often focus entirely or primarily on clinical mental disorders associated with IPV such as PTSD and depression. In contrast, practitioners and researchers with a holistic orientation that goes beyond mental disorders use the term “psychosocial” to capture the relational, psychological distresses (e.g., fear of losing one’s children) and non-clinical aspects (e.g., stigma and social exclusion) that go beyond mental disorders. Our contention is that this latter, wider usage of the term “psychosocial” is particularly appropriate in LMIC settings, most of which have a collectivist orientation that augurs against a primary focus on individual disorders. These contexts often have greater burdens of economic distress, material deprivation, and other forms of non-clinical distress than do the Western contexts that have been the dominant focus of research on IPV. This conceptual analysis and re-focusing has significant implications for practice. At present, psychological supports for IPV survivors frequently emphasize the treatment of mental disorders. Consistent with global guidelines on gender-based violence and mental health and psychosocial support  from humanitarian settings, our paper tries to make the case for more holistic approaches to supporting IPV survivors that address both mental disorders and the survivors’ main priorities such as economic well-being.

Reviewer 1’s call for collecting empirical data seems to be motivated by a desire to help the paper fit with the journal’s pre-defined categories of research papers or review papers. The call for adding new data to the paper, however, would not enable the paper to achieve its conceptual and analytic goal. Nor would it help to achieve its aims as a review, as discussed below.

In developing our paper, we chose to conduct an inductive, analytic review rather than a systematic review because of concerns about bias and wanting to establish a stronger foundation for the conduct of systematic reviews. A systematic review of psychosocial impacts of IPV would likely be biased by the fact that many health researchers use “psychosocial impact” to mean “mental disorders.” Even if efforts were made to reduce this bias by, for example, not reviewing the papers that focus on mental disorders, a systematic review is still not indicated since there is little agreement about which categories of “psychosocial impact” to look for. Much of the literature on IPV has come from Western settings, where more individualized concerns and approaches may be appropriate. Relatively little work has been done to identify the wider psychosocial impacts of IPV in a manner that reflects the lived experience of IPV survivors in LMIC settings. To do this, the paper takes an inductive approach based on narratives of IPV survivors and contextually rich descriptions from diverse LMIC settings. This inductive approach is useful in identifying psychosocial impacts that reflect the cultural and social differences and the challenges of living in LMIC settings. This inductive approach is particularly important since many of the effects of IPV on survivors are not discussed in the relevant papers as “psychosocial”. For example, a paper that examines how IPV increases family economic hardships may focus mostly on economics, but may  mention IPV survivors’ concerns about their children’s well-being and descriptions of how the women suffer as a result. Inductive reading of the papers is necessary for enabling a holistic identification of psychosocial impacts of IPV in LMIC settings. This identification helps to lay the foundation for quality systematic reviews in the future. We believe that the design of the paper as a combination of conceptual analysis and inductive review is both appropriate and necessary in this early stage of inquiry into IPV effects in LMIC settings.

Accumulation and Interaction: Reviewer 1 suggested that section 3.6 on accumulation and interaction be included in the Discussion section with more detail. We are grateful for this suggestion and have moved the material to the Discussion section and have added material on the longer term health impacts on IPV survivors and children.

Geographic focus: Reviewer 1 noted accurately that the research we have drawn upon comes mostly from LMIC in Asia and Africa. We have accepted their recommendation to clarify from the beginning that the examples come mainly from LMIC contexts in those geographic areas. Relevant revisions were made in statement of purpose in the Introduction section, and also in the Limitations section. We thank Reviewer 1 for these recommendations.

Reviewer 2 Report

Thank you for the opportunity to review this paper. It deals with a subject of high social relevance, by exploring from a gender-based, multidimensional perspective the psychosocial impact of IPV against women in low- and middle-income countries.

I have found this analytical study thorough and really interesting. The holistic approach the authors propose, although preliminary, opens a necessary broader frame for prevention, intervention and future research, that may contribute to fight this social scourge in the specific and diverse contexts of LMIC societies.

The text is well written, with a well-crafted structure. The rationale and arguments are supported by models and empirical studies, correctly referenced in the text. The introduction is adequate and the objectives and the analytical focus are both correctly posed at the end of it. The final discussion is consistent with the previous analysis, including limitations and future challenges.

There are only some minor comments that I would like to share with you, which hopefully you may find useful:

I would suggest including some key reference(s) that could help back up your statement in page 2, line 72-73 “the paper focuses on women and girls since they are most severely affected by IPV”, since this sentence pose the basis and main focus of the whole paper. I think your references to WHO (2021) and Devries et al. (2013) could work.

And only for your consideration, two ideas in order to clarify your focus on IPV against women (versus “bidirectional IPV”) throughout the paper:

- Consider to introduce, right after you have stated your main focus (page 2, line 75), the acronym IPVAW (“IPV Against Women”) instead of “IPV”, and to keep it hereon.

- Consider to bring this IPVAW-focus to the forefront of your paper, including this nuance also in the title.

In this sense, and again only for your consideration, maybe you could include a reference to this recent publication of Sardinha et al. (2022). I think it could fit with your focus:

Sardinha, L., Maheu-Giroux, M., Stöckl, H., Meyer,S. R., & García-Moreno, C. (2022). Global, regional, and national prevalence estimates of physical or sexual, or both, intimate partner violence against women in 2018. The Lancet, 399, 803-813. https://doi.org/10.1016/S0140-6736(21)02664-7

Some minor issues to review and address:

- In page 10, line 471, the reference to Stark et al.’s work is incomplete. Please, check.

- Please, explain the MHPSS acronym the first time it is mentioned (Page 11, line 515).

- In page 13, line 621 you cite HRW (2017), but I did not find the reference. Please, check.

- There seems to be a typo in page 15, line 710 (“are.also”) and in line 717 (“LAMIC”). Please, check.

Once again, thank you for your work on this important issue, and for the possibility to take part of it.

Author Response

Reviewer 2 offered valuable suggestions and we appreciated their support for our analytical approach. We have accepted all their suggestions, which are listed below point by point.

- We have included the suggested references needed to back up our statement that the paper focuses on women and girls since they are the most severely affected by IPV.

- On terminology, we have accepted their suggestions to use the acronym IPVAW (IPV Against Women) and include this also in the title of the paper. This approach helps to sharpen the focus of the paper and avoids overgeneralizing to (or tacitly discriminating against) other IPV affected people.

- We have included the Sardinha et al. (2022) reference—thanks for calling this valuable paper to our attention.

- We have checked and corrected the references to Stark’s work on p. 10.

- We have explained the MHPSS acronym on its first use on p. 11.

- We have clarified the Human Rights Watch (2017) citation and included the study in the references.

- We have corrected the spelling on LMIC on p. 15.

Round 2

Reviewer 1 Report

The clear delimitation of the concept to IPVAW and specifications together with adding the limitation section improved the document.